# Effects of raster terrain representation on GIS shortest path analysis

**F. Antonio Medrano** *

Department of Computing Sciences, Conrad Blucher Institute for Surveying and Science, Texas A&M University–Corpus Christi, Corpus Christi, Texas, United States of America

* antonio.medrano@tamucc.edu

## Abstract

Spatial analysis extracts meaning and insights from spatially referenced data, where the results are highly dependent on the quality of the data used and the manipulations on the data when preparing it for analysis. Users should understand the impacts that data representations may have on their results in order to prevent distortions in their outcomes. We study the consequences of two common data preparations when locating a linear feature performing shortest path analysis on raster terrain data: 1) the connectivity of the network generated by connecting raster cells to their neighbors, and 2) the range of the attribute scale for assigning costs. Such analysis is commonly used to locate transmission lines, where the results could have major implications on project cost and its environmental impact. Experiments in solving biobjective shortest paths show that results are highly dependent on the parameters of the data representations, with exceedingly variable results based on the choices made in reclassifying attributes and generating networks from the raster. Based on these outcomes, we outline recommendations for ensuring geographic information system (GIS) data representations maintain analysis results that are accurate and unbiased.

## Introduction

Spatial analysis is used to bring meaning and insights out of spatially referenced data, and the set of methods that are identified as spatial analysis tend to be some of the most heavily used in geographic information system (GIS) software [1]. As with any sort of analysis, the results from spatial analysis are highly dependent on the quality of the data provided, as well as the understanding that the GIS user has with respect to the methods used. A GIS user must almost always prepare and manipulate spatial data in order to make it suitable for use in analysis, and thus it is imperative that the user understand the impacts that these manipulations may have on the final results. Otherwise, the outcome of a spatial analysis may inadvertently be distorted. While misinformation through cartographic manipulations have been well documented [2, 3], if the GIS user has a desired outcome from the analysis they may even use data manipulations to covertly drive the solutions toward a desired goal. Thus, it is important to be aware of the

**Data Availability Statement:** https://doi.org/10.5281/zenodo.4540743 which is an archived snapshot of the following open source repository: https://github.com/antoniomedrano/pNISE.

**Funding:** Argonne National Laboratories, grant number 1F-32422 The funders had no role in study

design, data collection and analysis, decision to publish, or preparation of the manuscript.

**Competing interests:** The authors have declared that no competing interests exist.

effects of spatial data representation, and to establish guidelines that help to ensure that GIS analyses accurately represent real-world conditions and provide impartial solutions.

While GIS analysis techniques are numerous and broad, and an entire book could be written covering all impacts of data representation; the main objective of this article is to focus on the impacts of two common data transformations when representing terrain as a raster network for locating a linear feature using shortest path analysis: 1) defining the network generated by connecting raster cells to their neighbors, and 2) the range of the attribute scale that represents the costs to locate the feature at each raster cell. Raster-based shortest path analysis is the predominantly used method for locating linear features over terrain, such as new transmission line corridors [4–13], pipelines [14–16], roadways [17, 18], as well as analyzing the connectivity of a landscape for habitat analysis [19–23] and urban systems [24]. These applications typically require generating a set of non-inferior options that balance numerous competing interests such as economic cost, environmental impact, maintenance accessibility, visual pollution, etc.; and from that set of options a decision-making entity can select the final route alignment. Multi-objective shortest path (MOSP) analysis is commonly used for generating such alternatives, since it finds the set of optimal trade-off solutions between multiple competing objectives, and thus can find compromise solutions to best satisfy various parties with different values and priorities [9]. MOSP analysis is valuable at highlighting the representation effects of network connectivity and attribute scale since it provides a rich set of path solutions from which to see the effects of varying parameters of the data representations. When using just two objectives, MOSP analysis is known as biobjective shortest path (BOSP) analysis.

This study examines the effects of raster connectivity and attribute scale via BOSP analysis, comparing the number of Pareto-optimal solutions, the layout of the paths in decision space, and the performance of the solutions in objective space where applicable. We look at the guidelines found in the literature on locating transmission line corridors, and see how their recommendations affect the quality of the analytic solutions. In the discussion and conclusion, we provide guidelines to ensure that such spatial analyses are performed with the appropriate modeling accuracy and objectivity.

## Background

The world is infinitely complex and continuous, and modeling it exactly on a digital computer is not possible. Thus, representing space on a computer requires discretization of both space and attributes. In the context of corridor location, information on terrain is often generated from remote sensing digital imagery that consists of a regular grid of pixels, and thus raster is the natural representation of such spatial data. Spatial attribute information is categorized into one of four basic measurement levels: nominal, ordinal, interval, and ratio [25]. Some spatial analysis techniques can be performed on nominal and interval features, such as go/no-go suitability analysis, spatial overlay, or location set covering. But ratio-level data is required for shortest path analysis, location-allocation problems such as the $p$-Median problem [26, 27], or any other analysis that requires multiplying the attribute value by a distance. When a GIS analyst uses a dataset that contains nominal or ordinal level data such as landcover type, they will often have to perform a reclassification to convert it into ratio-scaled data. It is this conversion that can lead to erroneous or misleading results if the reclassification is performed carelessly, and in this article we examine the consequences of such inaccurate reclassification. The effects of representation on the results of spatial analysis have been a known problem with GIS for quite some time. Miller [28] observed that "Spatial analysis was mostly developed in an era when data was scarce and computational power was expensive. Consequently, traditional spatial analysis greatly simplifies its representations of geography". As technology progresses, he

suggests an ongoing "re-examination of geographic representation in spatial analysis". Tong and Murray [29] point out that "it is well recognized that findings can be highly dependent on how space is abstracted and represented. This can be due to the way we partition or conceptualize space" and that "much research is needed to reduce or alleviate errors and uncertainties in abstracting geographical space". This article addresses the impacts of these representation issues in the context of shortest path analysis, and provides recommendations for best practices to avoid such problems. Geospatial representation and its effects on analysis continues to be an active area of research. For example, Gaboardi and Folch [27] evaluated spatial network representation for allocating and connecting points on a network, and found this could have substantive effects on the results of a *p*-Median and *p*-Center location analysis.

Any shortest path computation requires a network upon which to find the least-cost route, as raster data sets do not fundamentally have a built-in network structure. Methods have been developed to convert a raster into a network by assuming that the center of each raster cell is a node and defining arcs as links that connect each cell to its neighboring cells. Each arc then has a cost function which is the distance-weighted sum of the attributes of cells the arc traverses, called the cost-distance function. If the arc has width then the area of the path that intersects a cell is typically used as a weight instead of the distance of the arc intersecting that cell [4, 30, 31]. Assuming zero width as is prevalent with most built-in GIS functionality, the objective cost for any path is thus the sum of these cost-distance weighted arcs that contiguously connect the origin and destination locations. A shortest path problem finds the path that minimizes this cost. In a multi-objective shortest path algorithm, each arc that makes up a path has multiple costs corresponding to each objective, and any path will have a set of objective scores where the scores represent the performance of the path with respect to each objective.

To convert a raster grid into a raster network, cells are most commonly connected to their neighbors according to a specified radius *R* (see Fig 1) [32]. *R* = 0 denotes connecting cells to their orthogonal neighbors (rook's move), *R* = 1 denotes connecting cells to their orthogonal and diagonal neighbors (queen's move), and *R* = 2 denotes additionally connecting cells via knight's moves. In a knight's move, the network arc spans two raster cells in one direction and one raster cell in the orthogonal direction, but it is considered a straight-line connection between the starting and ending points. The cost-distance for such an arc is a function attributes of the four raster cells it passes through multiplied by the total arc length [4]. The higher the radius used to generate the raster network, the less geometric distortion the network will

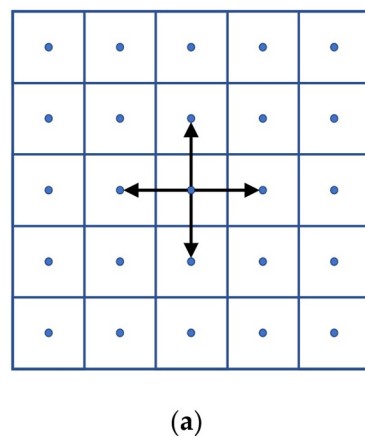
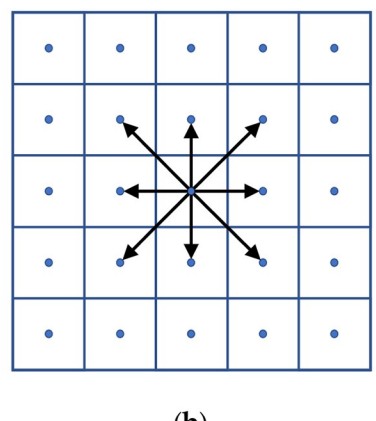
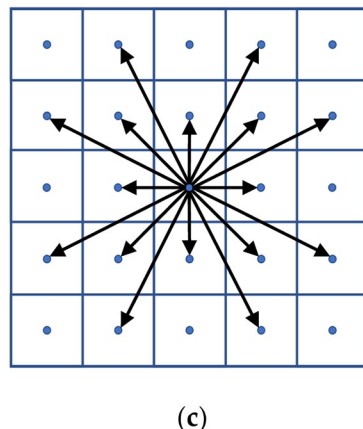

(a)                                             (b)                                             (c)

**Fig 1. Raster network connectivity (a) *R* = 0, (b) *R* = 1, (c) *R* = 2.**

have due to less restricted movement between cells; but this comes at the cost of a higher network density, resulting in longer computation times. Goodchild [32] calculated the worst-case geometric elongation for a shortest path when traversing a raster network of uniform cost, and found the $R = 0$ network imparts a 41.4% elongation error, the $R = 1$ network imparts an 8.2% elongation error, and the $R = 2$ network imparts a 2.79% elongation error. Higher radius values could be used to further reduce the elongation error, but Huber and Church [4] found that the $R = 2$ network provides the best trade-off between accuracy and computational burden on real geographic data. Elongation error is also encountered in the transportation literature as the *route factor*, defined as the ratio of the graph distance over the Euclidean distance between two points [33, 34].

Huber and Church [4] demonstrated that different radius-defined networks may result in optimal paths that take very different routes. This affects not just path objective costs but also real-world engineering design decisions, and is something our study examines within the context of bi-objective shortest paths. They also discuss raster orientation error, in which the optimal path length and route may also be subject to error due to the orientation of the raster network relative to the underlying topography. These results were confirmed by Antikainen [35], who found that with center connected paths the use of larger neighborhoods always yields better paths with less orientation error, at the expense of moderate increases in processing time. They also proposed an alternate boundary-based raster connectivity scheme, although we have yet to see their approach adopted in any GIS software. Seegmiller and Shirabe [31] propose an interesting method where in regions of constant raster cost (such as dense forest or water features or deserts), they define a start and end point within the monotonous region, and perform linear interpolation between the two to generate a corridor. This method enables much greater flexibility for path directions, but it is limited to straight-line paths within monotonous regions. Other publications that have looked at additional sources of error in raster network shortest path analysis include Huber [36] and Hong and Murray [37]; who found that varying raster cell size can have major implications on the objective value and route of a shortest path.

A paper recently published by Schito and Moncecchi [38] uses a very interesting and promising approach to generate their connectivity graph. They generate a bespoke connectivity graph for the particular origin and destination they select, and use complex geometric decision rules to connect the nodes with arcs. Like the experiments in this article, their method was beyond the capabilities of any existing GIS software, and thus they had to program their own with Python. While they are unable to share their code due to non-disclosure agreements, the paths they generate are free of geometric distortions and were deemed highly satisfactory by their stakeholders. If their code is ever released publicly, it would certainly warrant an examination with the methods used in this paper.

Transmission line corridor location affects many nearby people, all with different concerns and priorities. A proposed design must consider all stakeholder interests, which oftentimes contain conflicting priorities. For example, a utility company may want to build the cheapest power lines taking a straight-line path, whereas environmentalists may want the route to divert around a sensitive habitat. Multiple objectives are commonly encountered in such contentious public projects where the interests of diverse stakeholders must be considered when developing a set of alternatives for debate and decision-making. This is especially true for transmission line location, since these are often considered undesirable to locate near humans or wildlife [39–41]. These contentious design problems, affectionately known as wicked problems [42], may be subject to sneaky manipulations in order to guide the decision toward one party's desired outcome. This study uses biobjective shortest paths to shed light on the subtle techniques that GIS practitioners may use to accomplish such a desired outcome.

Because of the often-wicked nature of such problems, multi-objective optimization is commonly used for locating transmission lines. A multi-objective optimization problem entails finding the solutions that represent an optimal set of trade-off solutions between two or more objectives [24]. Aside from the methodologies we analyze, recent publications using multi-objective shortest paths to locate transmission lines over terrain raster data include [9–12]. All of these recent publications contain results with geometric distortions caused by the limitations on raster network connectivity in the GIS software they used, effects that we examine in this article.

Biobjective shortest path solutions are visualized and evaluated in both decision space and objective space (see Fig 2), where decision space is the real-world cartographic representation of the region where the path is being placed, and objective space depicts how that path performs with regards to each objective in comparison to other paths. Paths are linear features in decision space, and have corresponding point features in objective space (three paths are depicted in Fig 2a and the performance of those three paths are highlighted in Fig 2b). The set of non-dominated or Pareto-optimal solutions are those where there does not exist any other feasible solution that performs better in all objectives. These solutions form the trade-off frontier, which in the case presented in Fig 2 involves both minimizing cost and minimizing environmental impact.

Supported non-dominated solutions consist of the convex set of Pareto-optimal solutions and can be computed by solving single-objective problems combining the multiple-objectives via carefully selected weights [43]. Un-supported non-dominated solutions are the Pareto-optimal solutions that are not part of the convex frontier, and require specialized multi-objective algorithms to compute. Finding the set of all supported non-dominated solutions is computationally weakly polynomial, while computing the unsupported solutions is NP-Hard [44]. This study considers only the supported non-dominated solutions, as they provide a

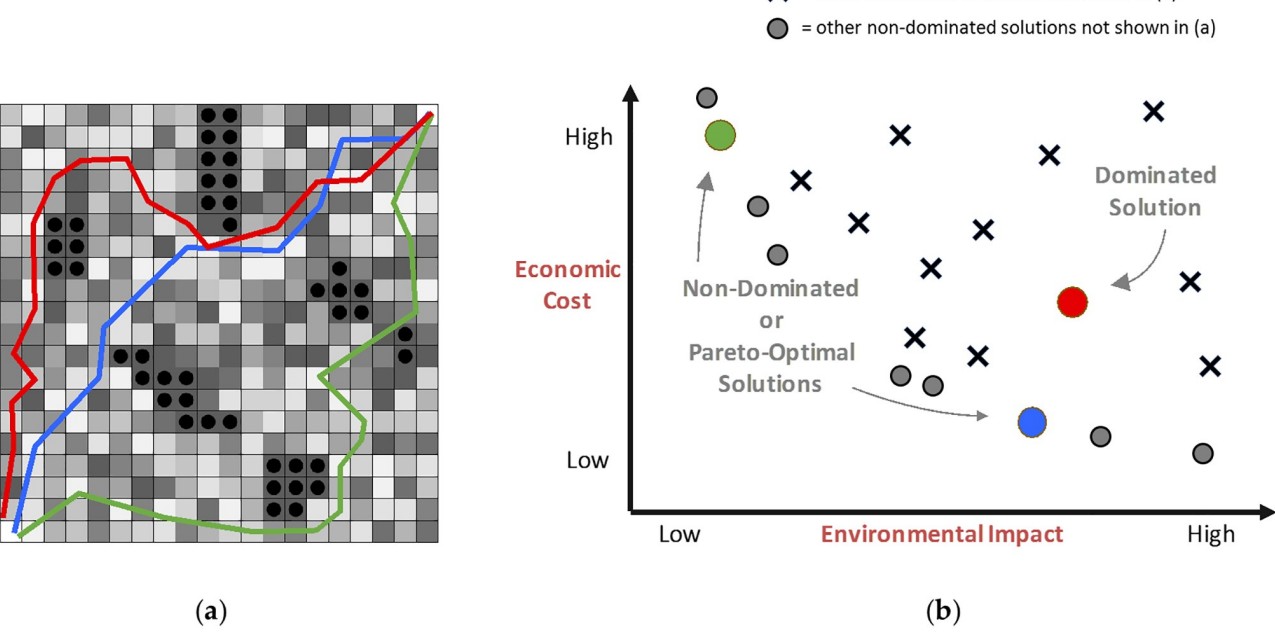

**(a)**  **(b)**

**Fig 2. Evaluating three paths in both (a) decision space, and (b) objective space.**

sufficiently rich solution set for demonstrating terrain raster representation errors with shortest path analysis.

## Materials and methods

### Data

The analysis in this study used GIS raster data sets assembled and used by the Eastern Interconnection States' Planning Council (EISPC). These data sets are intended to facilitate the identification of potential energy sites and transmission line corridors within the EISPC region, which spans 39 eastern US states, Washington D.C. and 8 Canadian provinces. The data was assembled jointly by Argonne National Laboratory, Oak Ridge National Laboratory and the National Renewable Energy Laboratory as a part of their EISPC Energy Zones Study (EZS) [45].

The EZS data contains numerous geographical information layers that would be used in a suitability analysis for locating new energy infrastructure, and is available through the EISPC Energy Zones Mapping Tool (ezmt.anl.gov). As of December 2020, the EZS contained 332 data layers, including land cover type, slope, water bodies, watersheds, essential habitats, earthquake intensities, existing transmission lines, substations, rail and roadways, just to name a few. Our study uses a 1000×1000 raster subset of the EZS data, with a 250 meter cell size centered at 36.516° N, 88.687° W. The region analyzed is in the Kentucky Lake region where the Tennessee River and the Cumberland River intersect the Ohio River, and includes portions of Tennessee, Kentucky, Illinois and Missouri. All maps in this article were created using the EZS numerical data, rendered programmatically with Java and the Processing API. No GIS software was used, and all code and data used to generate the maps is contained in the public Github repository [46]. All maps are oriented with North as up, and at this scale all maps in this article have an extent of 250km × 250km.

The EZS raster data was used to create two cost surfaces for a bi-objective optimization, where the competing objectives were to minimize 1) the infrastructure construction cost, and 2) the environmental impact. Since these objectives are not explicitly in the EZS data set, it was necessary to derive ratio-scale cost surfaces from the available layers. The slope layer, in percent slope, was used to develop a construction cost surface. The land cover type layer, categorized according to the National Land Cover Database 2016 (NLCD2016) which was publicly released in May 2019 [47, 48], was used to generate an environmental impact cost surface. The slope and land category attributes were then converted to ratio-scaled cell costs according to the terrain cost multipliers recommended by the Western Electricity Coordinating Council (WECC) [7], listed in Table 1. In all experiments, all cost-surfaces were scaled to equal ranges between the two objectives.

Fig 3 graphically displays the EISPC data maps used in the analysis, represented as the raw 1000×1000 rasters for (a) land use type and (b) slope, and then reclassified as (c) environmental impact and (d) economic cost. All maps in Fig 3 were created using Java for this study, but Fig 3a uses the same colors as the NLCD2016 class legend (https://www.mrlc.gov/sites/default/files/NLCD_Colour_Classification_Update.jpg), and the other maps in Fig 3 use light colors represent low slope or cost, and dark colors to represent high slope or cost according to the classifications in Table 1. Note that in cost layers derived from the land cover layer (Fig 3c), rivers and lakes have a high cost because it is expensive to build over water. In costs derived from the slope layers (Fig 3d), water features have a low cost because water is represented as flat. In a real-world transmission line location analysis, water would likely have a high cost with respect to both environmental impact and monetary cost. The WECC classifications were developed as single-objective cost multipliers where a high cost in one measure would carry

**Table 1. Attribute reclassification for fixed $C_{min}$ and varying amplitude.**

| NLCD2016Value | NLCD2016 Feature | WECC Feature | WECCValue | [1,2] | [1,5] | [1,10] | [1,20] | [1,50] | [1,100] |
|---|---|---|---|---|---|---|---|---|---|
| 11 | open water | n/a | 3.250 | 2.000 | 5.000 | 10.000 | 20.000 | 50.000 | 100.000 |
| 21 | developed, open space | suburban | 1.270 | 1.120 | 1.480 | 2.080 | 3.280 | 6.880 | 12.880 |
| 22 | developed, low intensity | suburban | 1.270 | 1.120 | 1.480 | 2.080 | 3.280 | 6.880 | 12.880 |
| 23 | developed, medium intensity | urban | 1.590 | 1.262 | 2.049 | 3.360 | 5.982 | 13.849 | 26.960 |
| 24 | developed, high intensity | urban | 1.590 | 1.262 | 2.049 | 3.360 | 5.982 | 13.849 | 26.960 |
| 31 | barren land (rock/sand/clay) | scrub/flat | 1.000 | 1.000 | 1.000 | 1.000 | 1.000 | 1.000 | 1.000 |
| 41 | deciduous forest | forested | 2.250 | 1.556 | 3.222 | 6.000 | 11.556 | 28.222 | 56.000 |
| 42 | evergreen forest | forested | 2.250 | 1.556 | 3.222 | 6.000 | 11.556 | 28.222 | 56.000 |
| 43 | mixed forest | forested | 2.250 | 1.556 | 3.222 | 6.000 | 11.556 | 28.222 | 56.000 |
| 52 | shrub/scrub | scrub/flat | 1.000 | 1.000 | 1.000 | 1.000 | 1.000 | 1.000 | 1.000 |
| 71 | grassland/herbaceous | scrub/flat | 1.000 | 1.000 | 1.000 | 1.000 | 1.000 | 1.000 | 1.000 |
| 81 | pasture/hay | farmland | 1.000 | 1.000 | 1.000 | 1.000 | 1.000 | 1.000 | 1.000 |
| 82 | cultivated crops | farmland | 1.000 | 1.000 | 1.000 | 1.000 | 1.000 | 1.000 | 1.000 |
| 90 | woody wetlands | wetland | 1.200 | 1.089 | 1.356 | 1.800 | 2.689 | 5.356 | 9.800 |
| 95 | herbaceous wetlands | wetland | 1.200 | 1.089 | 1.356 | 1.800 | 2.689 | 5.356 | 9.800 |
| | **Slope** | **WECC Feature** | **WECC Value** | **[1,2]** | **[1,5]** | **[1,10]** | **[1,20]** | **[1,50]** | **[1,100]** |
| | < 2% | flat | 1.000 | 1.000 | 1.000 | 1.000 | 1.000 | 1.000 | 1.000 |
| | 2–8% | rolling hill | 1.300 | 1.600 | 3.400 | 6.400 | 12.400 | 30.400 | 60.400 |
| | > 8% | mountain | 1.500 | 2.000 | 5.000 | 10.000 | 20.000 | 50.000 | 100.000 |

over to the overall composite cost. Rather than divert from the published WECC values, we chose to keep them since for all other attributes they provide a very good ratio-scaled mapping to objective costs and it does not affect the overall evaluation of terrain network representation parameters. But it is important to note that any real-world analysis should develop custom application-specific costs to map the attributes to the modeled objectives.

## Algorithms

This analysis implemented the parallel bi-objective shortest path algorithm described in Medrano and Church [49] to compute the complete set of supported (convex) non-dominated path solutions using an origin at the lower-left corner of the raster region, and a destination at the top-right corner. This algorithm, called pNISE is a parallel implementation of the NISE algorithm commonly used to find the supported solutions of biobjective network optimization problems [23]. The algorithm is efficient at computing the Pareto-optimal path sets for biobjective shortest path problems of reasonably large graph size, which in the case of the $R = 2$ network contains 1 million nodes and approximately 16 million arcs. All code was written in Java, and visual results were rendered using the Processing API (processing.org). The reader is invited to download both the Java and the Processing codes from Github [46]. Coding a custom geospatial analysis tool rather than depending on existing GIS software allowed for exploring capabilities beyond those built-in to existing GIS tools. By evaluating if there are benefits to expanding how GIS software represents raster terrain as a network, we can make recommendations for features that should be added to GIS software.

## Raster network connectivity

Huber and Church [4] previously examined the effects of network connectivity on single-objective shortest paths on fabricated data, finding that altering the connectivities resulted in

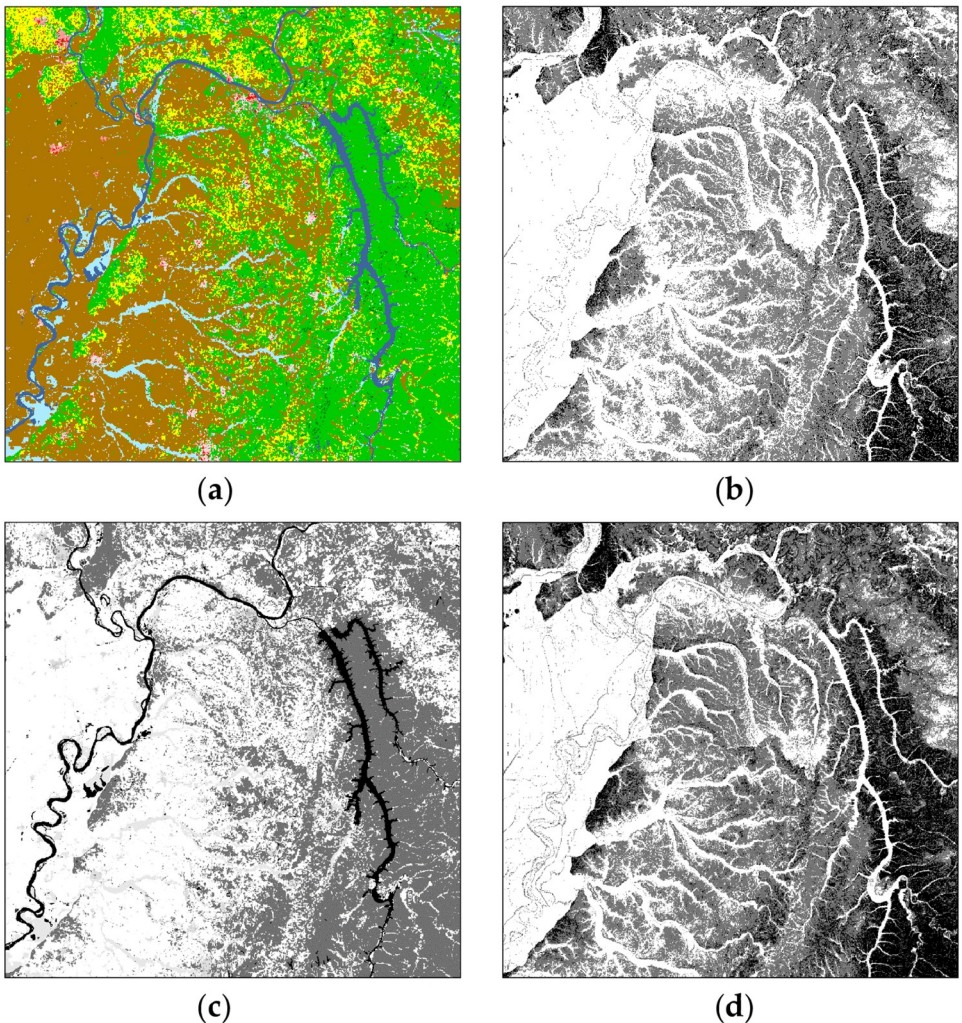

**Fig 3. 1000×1000 EISPC raster data (a) land cover (b) slope (c) environmental impact (d) economic cost.** In (b) light color is less slope and dark color is more slope, and in (c) and (d) dark color is high cost and light color is low cost.

differences in both path-objective performance and path topologies. In this section we perform a similar analysis with varying the network connectivity, and with biobjective shortest paths on the much larger EISPC real-world raster data. With the different connectivities we analyze in objective space the objective values of the Pareto-optimal path set in objective space and the number of paths that compose the complete convex Pareto-optimal path set. Qualitatively we compare the effects on path topologies in decision space, examining if the analyses exhibit different geometric distortions due to the parameters we vary when run on the same data. This multi-objective approach allows us to test the impacts of network connectivity on path delineation and performance on multiple network topologies via multiple weightings of the underlying raster layers, rather than just one single raster network used in previous studies.

Fig 4 displays the complete set of non-dominated paths with the South-West corner as the origin and the North-East corner as the destination, using networks with $R = 0$, 1, and 2. All use the WECC attributes scaled to [1, 10] for both objectives. What is immediately noticeable are the geometric artifacts for each type of network connectivity in the region west of the river

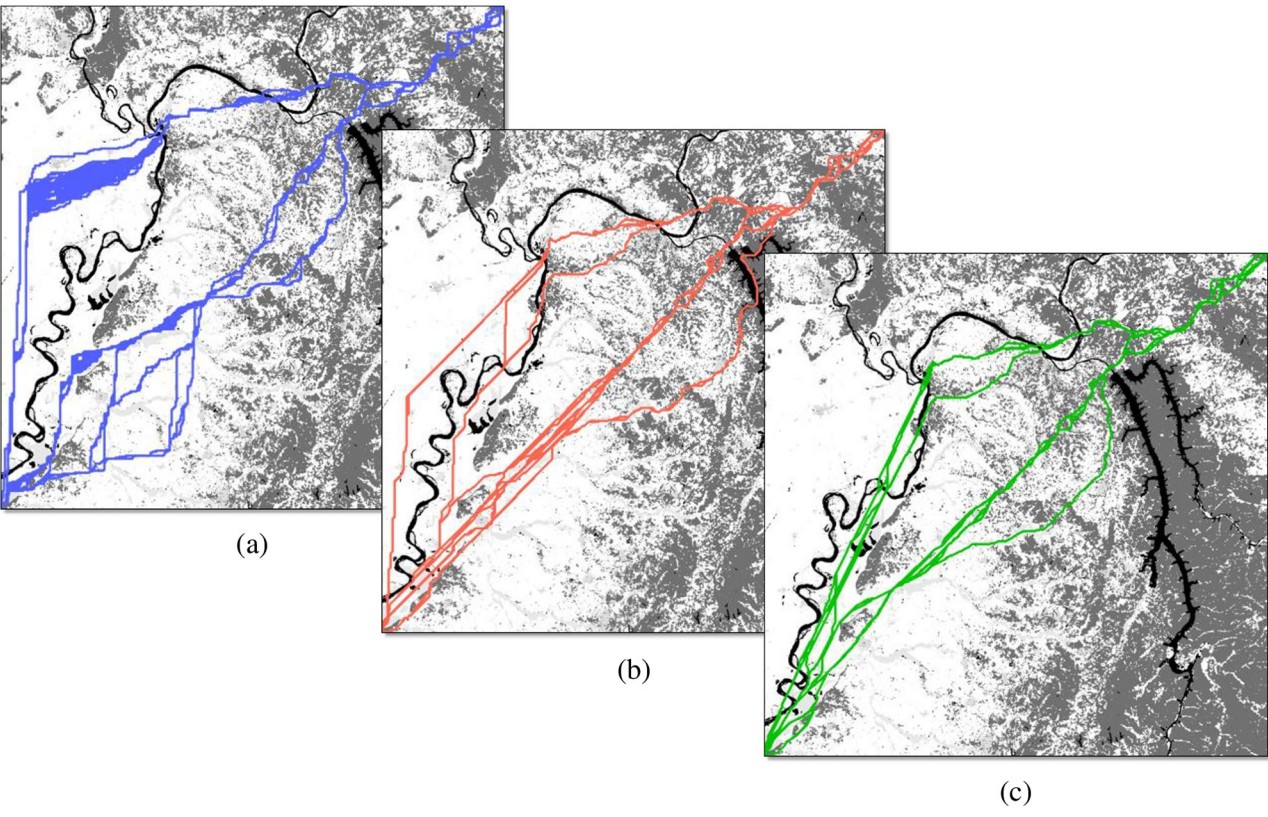

**Fig 4. Non-dominated solutions in decision space for (a)** $R = 0$ **(blue); (b)** $R = 1$ **(red); and (c)** $R = 2$ **(green).**

with relatively constant cost in both objectives. In this region, the $R = 0$ paths have a strong tendency to traverse either vertically or to alternate between vertical and horizontal arcs in order to approximate a diagonal traversal. The alternating artifacts are a clear indicator of an orientation error of the type illustrated in Huber and Church [4]. The $R = 1$ paths display clear regions of vertical or diagonal routes, aligned with the restrictions imposed by the available arc directions. The $R = 2$ paths do sometimes align with knight's move directions, but overall display the least amount of visible geometric distortion due to having the fewest route alignment restrictions. While discretizing continuous space will always result in some level of geometric distortion, it is clear that $R = 2$ connectivity dramatically reduces geometric distortion with minimal additional complexity as compared to the $\underline{R} = 1$ connectivity that most GIS software currently uses.

Fig 5 shows the supported, Pareto-optimal paths for all three connectivities in objective space. The $R = 0$ Pareto set (blue) contains 88 distinct paths, the $R = 1$ Pareto set (red) contains 145 paths, and the $R = 2$ Pareto set (green) contains 270 paths. There are major differences in the location of the Pareto-frontiers in objective space: as the network connectivities increase the objective scores decrease, and are in agreement with the theory developed in Goodchild [32] and previous experiments in Huber and Church [4]. The most pronounced difference is in going from $R = 0$ to $R = 1$, but there is still a distinct difference also between $R = 1$ and $R = 2$. If using the objective values to calculate expected costs of multi-million-dollar projects, a three to four percent increase in path length can mean significant errors in the budgetary estimates of potential alternatives. Presumably in the interest of minimizing computation time to determine optimal routes with older hardware, common GIS software packages do not

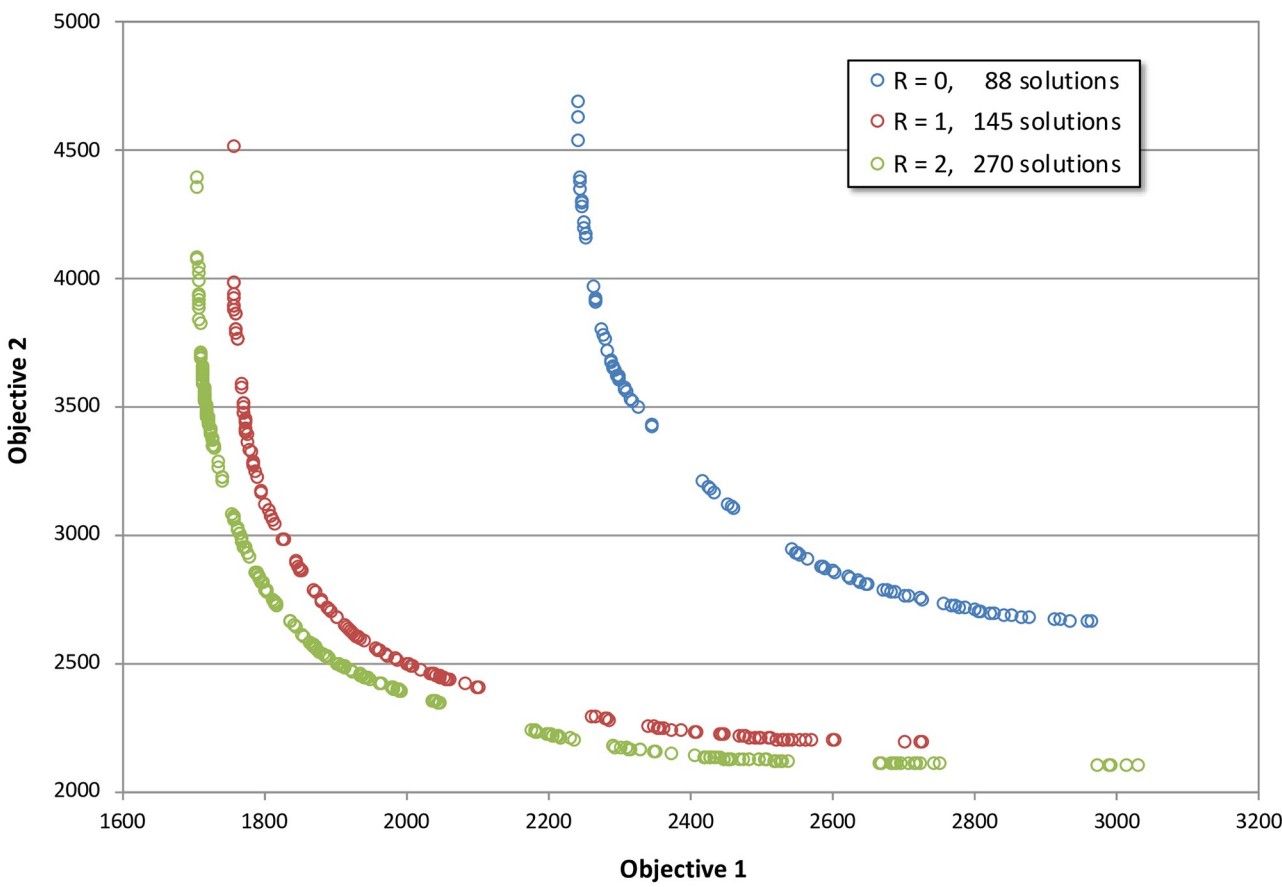

**Fig 5. Non-dominated solutions in objective space for $R = 0$ (blue), $R = 1$ (red), and $R = 2$ (green).**

include $R = 2$ network connectivity as an option; this capability currently has to be manually scripted into the analysis. Since $R = 2$ shortest path computation has become trivial for most real-world data sets using modern computing hardware, GIS software makers absolutely should incorporate an option to generate $R = 2$ networks as a built-in functionality.

## Attribute scale classification

A GIS practitioner will often need to reclassify attributes in order to prepare spatial data for analysis. Any shortest path analysis requires ratio-scaled data since a path cost is calculated as the sum of products of the attribute values and distances; the calculation is not associative. In other words, one cannot add a constant value to the costs of all nodes in a raster network and expect to get the same shortest path result. By adding a constant value, the shortest path algorithm will then be biased more towards finding a path that minimizes the number of arcs rather than the path of combined least impact with respect to the objectives. Thus, if an analysis is to be performed on land cover type and slope raster raw data, but decision-makers are actually trying to measure environmental impact and economic cost for locating the feature, then the data requires an attribute value conversion. Past literature for transmission line location has used a variety of approaches for these conversions:

1.  The Georgia Transmission Corporation [5]

    •  Scale all costs from 1 to 9 for all feature layers

- No mention that scaling should reflect actual costs

2. Bagli and Geneletti [6]

- All costs scaled from 0 to 1
- No mention that scaling should reflect actual costs

3. Western Electricity Coordinating Council (WECC) [7]

- Costs/mile for different kinds of transmission lines
- Costs/acre to purchase or lease land
- Cost multipliers for terrain type and slope

4. Esri cost surface online tutorials [50, 51]

- All costs scaled from 1 to 10
- No mention that scaling should reflect actual costs

Approaches 1 and 2 were made for performing multi-objective shortest path analysis on raster terrain networks, and are problematic because they use arbitrary ranges for the cost values that have no real-world meaning for path performance. Approach 3 is intended for cost estimation of a potential route, and does use true ratio-scaled cost multipliers to reclassify data according to slope or land use. The output of this approach gives results in actual dollar values for each route analyzed. Approach 4 is intended as a how-to online tutorial and casually scales everything from 1 to 10, and makes no mention that attributes should in-fact be scaled to actual real-world costs.

The geospatial analyses in this article demonstrate what can go wrong when cost ranges are picked arbitrarily without any correlation to actual costs. The analysis in the next section varies the amplitude while maintaining the same minimum cost and relative classification breaks. For example, suppose you have to reclassify features that are deemed as low, medium and high environmental impact into ratio scaled data. One could assign a cost of 1 to low-impact cells, a cost of 2 to medium-impact cells, and a cost of 4 to high-impact cells. We define the minimum cost as $C_{min}$, the maximum cost as $C_{max}$, and shorthand for reclassification range as [$C_{min}$, $C_{max}$]. We define the amplitude as $C_{max} - C_{min}$. In the above example, $C_{min} = 1$, $C_{max} = 4$, the range is [1, 4], and the amplitude is 3. Now suppose we want to double the amplitude to 6 but maintain the same minimum cost, then low-impact would cost 1, medium-impact would cost 3, and high-impact would cost 7, i.e. [1, 7]. Overall, this is equivalent to marking the classification breaks on a rubber band, then anchoring the lower bound and stretching the upper bound, as shown in Fig 6a.

The analysis in the following section varies the minimum cost for the reclassification while maintaining the same amplitude and relative classification breaks, as shown in Fig 6b. A [2, 5] shift would have a cost of 2 for low-impact, 3 for medium-impact, and 5 for high-impact, effectively adding 1 to every value as compared to a [1, 4] classification.

In both experiments, the same underlying data is used using the exact same relative interval proportions, while varying only the amplitudes or the minimum costs. In other words, all experiments use the same WECC feature costs for each land category or slope, but the costs are then stretched according to the amplitude or are shifted by the minimum cost value, as shown graphically in Fig 6. All classification experiments here use the same $R = 2$ network connectivity.

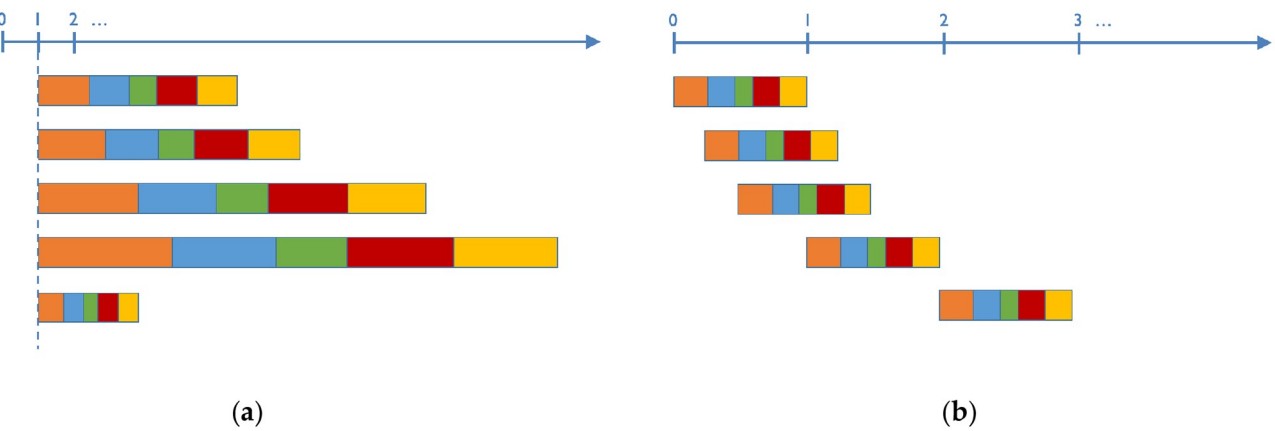

**Fig 6. Attribute classification modification via (a) stretching, or (b) shifting.**

Both experiments yield results allowing for both quantitative and qualitative comparisons. When varying the range and amplitude of the raster attributes, one cannot directly compare the objective values of the results. The key analysis is the qualitative comparison of how variations in the attribute ranges affect the path topologies in decision space. The number of paths in the Pareto-optimal path set can be compared quantitatively as well. Combined, these two measures indicate how choices made in selecting the attribute ranges affect the character and diversity of the resulting optimal path alternatives.

Let us informally define the dynamic range of a reclassification scale as the following, as this is a useful measure to compare the effects of the reclassification schemes:

$$Dynamic\ Range = \frac{maximum\ attribute\ cost}{minimum\ attribute\ cost} = \frac{C_{max}}{C_{min}} \qquad (1)$$

In Fig 6a, the shorter bars represent reclassifications with small dynamic range, and the longer bars with large dynamic range. In Fig 6b, the reclassifications on the left (close to zero) will have large dynamic ranges due to the smaller denominator, and those on the right will have smaller dynamic ranges.

## Reclassification: Varying the amplitude

This experiment maintained $C_{min} = 1$ while changing the attribute scale amplitude. Attribute reclassification values are shown in Table 1, varying $C_{max}$ from 2 to 100. Land use features were used for one objective layer, and the slope was used for the other objective layer. All experiments used the equal ranges for the two objectives in order to maintain equal emphasis between them.

Fig 7 displays the results of this analysis in decision space. Low amplitude small dynamic range solutions tended toward straight paths that are approximated by a Euclidean shortest path, while high amplitude large dynamic range solutions tended to have greater deviations and spatial diversity. Recalling that arc costs are a product of the arc distance and the cell attribute values, it is clear that varying the ranges in this manner results in a trade-off between minimizing the spatial length of a shortest path, i.e. the Euclidean tendency, and the need to avoid cells with high cost attributes. As attribute amplitudes increase, the total number of non-dominated path solutions increase as well. This, too, is an indicator of the spatial vs. attribute trade-off, as the extreme and unrealistic case of a homogenous flat-cost geographic space would have a single non-dominated solution consisting of the Euclidean shortest path.

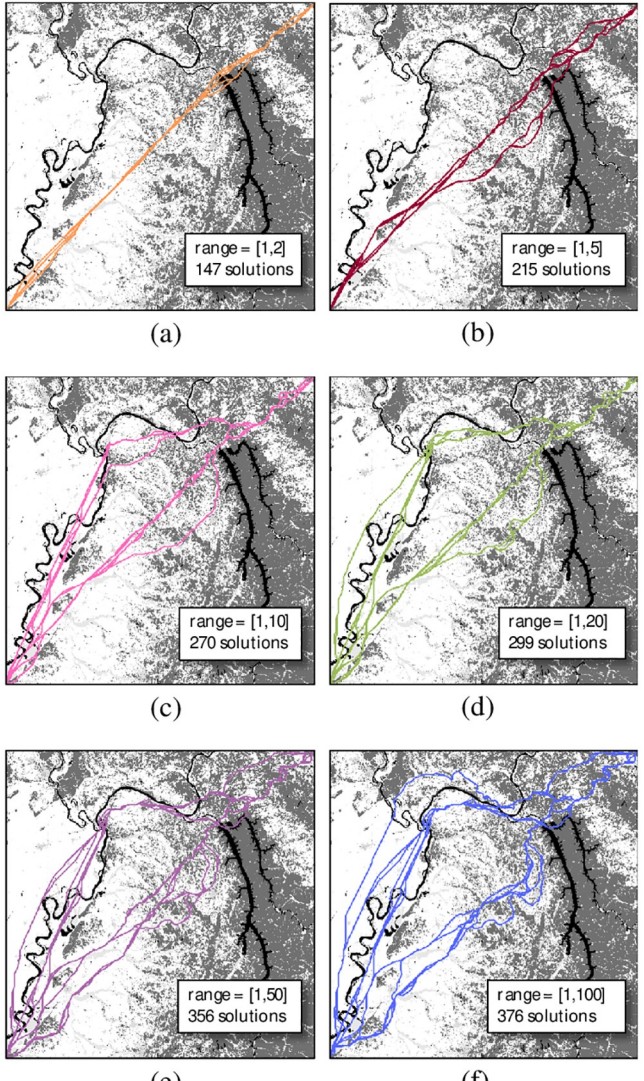

**Fig 7. Decision space solutions for a constant $C_{\min}$ while varying the amplitude to the following ranges: (a) [1,2]; (b) [1,5]; (c) [1,10]; (d) [1,20]; (e) [1,50]; and (f) [1,100].**

## Reclassification: Varying the minimum value

This experiment maintained a constant amplitude of 1 while varying the value of Cmin. Attribute reclassification values are shown in Table 2, varying $C_{\min}$ from 0 to 5. The land use features were used for one objective layer, and the slope was used for the other objective layer. All experiments used the equal ranges for the two objectives in order to maintain equal emphasis between them.

Fig 8 displays the results of this analysis in decision space using the shifted attribute scales all with an amplitude of 1. What is immediately noticeable is the extreme behavior of the [0,1] range. In the western area with homogenous regions of zero cost, the paths appear to wander aimlessly in a sort of Brownian motion. The zero cost cell attributes mean that arc costs in this region are also zero, and there is no penalty for taking a long and windy path. Thus, the path generated by Dijkstra's algorithm is subject to the pseudo-random motion that comes from

**Table 2. Attribute reclassification for fixed amplitude and varying $C_{min}$.**

| NLCD2016Value | NLCD2016 Feature | WECC Feature | WECC Value | [0,1] | [0.1,1.1] | [0.2,1.2] | [1,2] | [2,3] | [5,6] |
|---|---|---|---|---|---|---|---|---|---|
| 11 | open water | n/a | 3.250 | 1.000 | 1.100 | 1.200 | 2.000 | 3.000 | 6.000 |
| 21 | developed, open space | suburban | 1.270 | 0.120 | 0.220 | 0.320 | 1.120 | 2.120 | 5.120 |
| 22 | developed, low intensity | suburban | 1.270 | 0.120 | 0.220 | 0.320 | 1.120 | 2.120 | 5.120 |
| 23 | developed, medium intensity | urban | 1.590 | 0.262 | 0.362 | 0.462 | 1.262 | 2.262 | 5.262 |
| 24 | developed, high intensity | urban | 1.590 | 0.262 | 0.362 | 0.462 | 1.262 | 2.262 | 5.262 |
| 31 | barren land (rock/sand/clay) | scrub/flat | 1.000 | 0.000 | 0.100 | 0.200 | 1.000 | 2.000 | 5.000 |
| 41 | deciduous forest | forested | 2.250 | 0.556 | 0.656 | 0.756 | 1.556 | 2.556 | 5.556 |
| 42 | evergreen forest | forested | 2.250 | 0.556 | 0.656 | 0.756 | 1.556 | 2.556 | 5.556 |
| 43 | mixed forest | forested | 2.250 | 0.556 | 0.656 | 0.756 | 1.556 | 2.556 | 5.556 |
| 52 | shrub/scrub | scrub/flat | 1.000 | 0.000 | 0.100 | 0.200 | 1.000 | 2.000 | 5.000 |
| 71 | grassland/herbaceous | scrub/flat | 1.000 | 0.000 | 0.100 | 0.200 | 1.000 | 2.000 | 5.000 |
| 81 | pasture/hay | farmland | 1.000 | 0.000 | 0.100 | 0.200 | 1.000 | 2.000 | 5.000 |
| 82 | cultivated crops | farmland | 1.000 | 0.000 | 0.100 | 0.200 | 1.000 | 2.000 | 5.000 |
| 90 | woody wetlands | wetland | 1.200 | 0.089 | 0.189 | 0.289 | 1.089 | 2.089 | 5.089 |
| 95 | herbaceous wetlands | wetland | 1.200 | 0.089 | 0.189 | 0.289 | 1.089 | 2.089 | 5.089 |
| | **Slope** | **WECC Feature** | **WECC Value** | [0,1] | [0.1,1.1] | [0.2,1.2] | [1,2] | [2,3] | [5,6] |
| | < 2% | flat | 1.000 | 0.000 | 0.100 | 0.200 | 1.000 | 2.000 | 5.000 |
| | 2–8% | rolling hill | 1.300 | 0.600 | 0.700 | 0.800 | 1.600 | 2.600 | 5.600 |
| | > 8% | mountain | 1.500 | 1.000 | 1.100 | 1.200 | 2.000 | 3.000 | 6.000 |

the tie-breaking rules that were used in the particular implementation. In terms of dynamic range, zero costs represent a denominator of zero, which results in an undefined ratio. These model results are clearly unrealistic to any real-world behaviors, and as such, zero-cost attributes should always be avoided. Slightly above the [0,1] range, the solutions were diverse and mostly influenced by the attributes. This represents a large dynamic range due to the small denominator in the minimum attribute costs. As the ranges continue to shift higher, i.e. smaller dynamic range, the paths become more Euclidean as the arc costs gradually emphasize geometry over attribute values. The attribute cost values increase, but the relative dynamic range ratios between high and low costs become minimal. With regards to the number of solutions, the higher the range is shifted the fewer non-dominated solutions that exist; with the exception of the pathological [0,1] case, which had far fewer solutions than the slightly higher [0.1, 1.1] range.

## Discussion: The impacts of dynamic range

The experiments here found that a lower dynamic range will bias results toward Euclidian shortest paths, with fewer and less-diverse solutions. A higher dynamic range will bias results that emphasize minimizing objective costs, with more solutions of greater spatial diversity. Even in regions that appear relatively homogenous on the map, if the selected attribute scale has a large dynamic range then the resulting paths will include large deviations to avoid regions of slightly higher cost (see Figs 7 and 8). And in the case where $C_{min} = 0$ the dynamic range is undefined; and results showed that such a reclassification scheme to be problematic with path solutions that were random and unrealistic, and should be avoided at all costs. Because these results are derived from analysis on numerous raster terrain networks via the various weightings between the two objectives, as opposed to previous literature that only looked at one single-objective network, it is safe to say that these path deviation correlations with the dynamic range are more general than those from the previous literature.

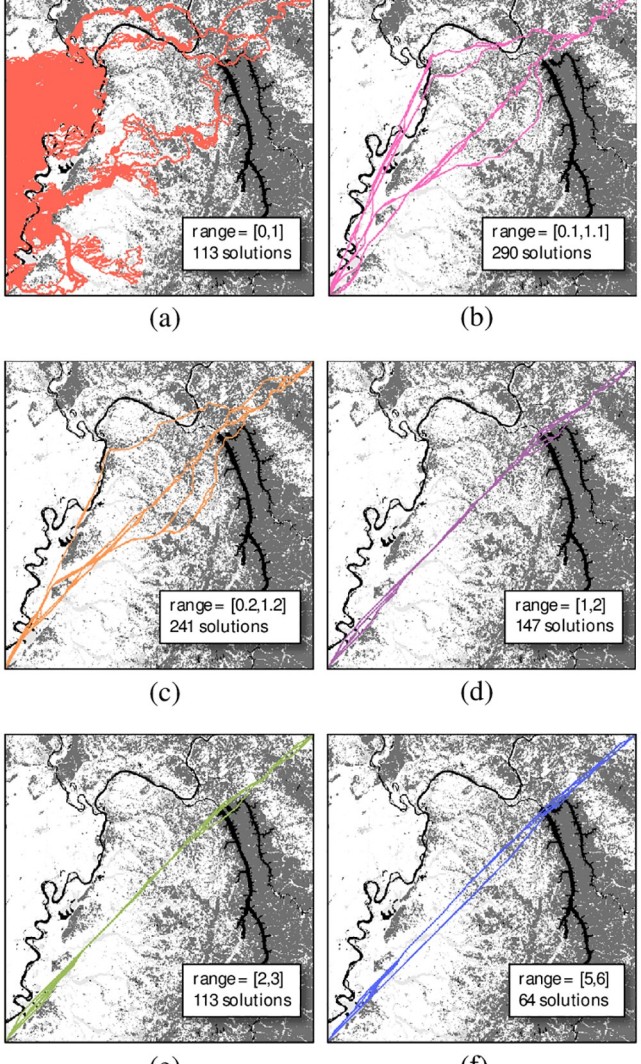

**Fig 8. Decision space solutions for a constant amplitude while varying C$_{min}$ to the following ranges: (a) [0,1]; (b) [0.1,1.1]; (c) [0.2,1.2]; (d) [1,2]; (e) [2,3]; and (f) [5,6].**

While in this article we display the classification results for $R = 2$ networks only, we observed similar trends for $R = 1$ and $R = 0$ connectivities as well. The Github repository [46] contains a folder with screen captures for the analysis results of all combinations of $R$-values and classifications mentioned in this article, and we invite the reader to review them.

The variability of results as a function of the attribute scales is why it is imperative that reclassification costs are assigned as ratio-scaled values based on real-world metrics, and not arbitrary interval-scaled values. This might seem obvious, but for a spatially unaware user learning how to perform this analysis, three out of the four methodologies cited earlier make no mention of scaling to real-world costs and instead instruct users to scale to arbitrary cost ranges. Calculating ratio-scaled costs are simple for tangible expenses such as the economic cost to locate in a particular cell, and is done quite well in the WECC cost estimation guide-lines. But for somewhat intangible costs such as environmental impact or maintenance accessi-bility the process is less clear. Questionnaires [52] or approaches like that of AHP [53] should

be implemented to develop true ratio scales for such intangible costs, so that there is supporting evidence to justify the relative cost-ratios despite the subjective nature of the criteria.

## Conclusions

Normative spatial analysis is used when decisions must be made on how to spatially configure a new design in order to maximize its utility or minimize its cost. The process of performing this analysis requires first generating a model from available data that reflects the conditions within the regional extent and relevant design factors. When designing projects to be located on natural landscapes, raster data is often the most appropriate; but as with any model, there are many considerations that must be made to ensure model accuracy. In this study, we have examined the errors and distortions associated with shortest path analysis when using raster representation of terrain, and how decisions made in the data preparation stage affected the results of the analysis.

First, we examined the effects of raster network connectivity on biobjective shortest path analysis, and found that the number of solutions, the spatial configurations of the routes, and the objective values of the Pareto-optimal set were all affected by the choice of the network connectivity used. While most popular GIS software packages only provide the ability to run analyses on $R = 0$ and $R = 1$ networks, it was found that $R = 2$ networks provided more alternatives, with less orientation error, and that their solutions consistently had lower objective costs than the $R = 1$ network paths. Given that continuing advances in computational power make shortest path analysis a trivial task for most common data, we unequivocally believe it is long overdue for major GIS software companies to add the built-in ability to generate $R = 2$ networks for spatial analysis.

Next, we examined the effects of reclassification to convert raw data features into cost surface rasters. We defined the dynamic range as being the ratio of the maximum cost divided by the minimum cost. We then ran the same analysis with the same relative interval breaks between different attribute costs, varying only the range of the attribute cost mappings. The purpose of this experiment was to demonstrate that selecting arbitrary cost ranges, such as from 1 to 5 or 9 or 10, will significantly impact the results. We found that lower dynamic range reclassifications resulted in fewer path solutions that tended toward straight-line Euclidean shortest paths, while higher dynamic range reclassifications tended toward more path solutions that emphasized avoiding high attribute costs more than geometric factors. If an analyst has a motivation or preference for a more Euclidean solution, they could shift the classification range to a lower dynamic range to achieve this while still giving the appearance of a truly objective analysis. It should always be emphasized in all methodologies and tutorials that in order to maintain complete objectivity, ratio-scaled reclassifications should always be used: via direct conversion for tangible costs such as construction costs, or via surveys of relevant parties to gauge the proportional attribute impacts. These considerations are completely missing in all but one of the cited tutorials, resulting in methodologies unrepresentative of real-world conditions. Constituents should also inquire about the methods used in such an analysis when presented with alternatives developed by an entity who may have their own motives.

The geographic world is infinitely complex, and no model can perfectly capture every nuance of the spatial features that will affect the outcome of a normative spatial analysis. Approximations must be made in order to represent the world in a manner that can be computed upon, and these approximations will always come with sources of error and distortions. But it is important to be aware of these representation errors, and to use the best practices outlined here to mitigate their effects on the analysis so that the model can best reflect accurate real-world criteria and result in objectively unbiased solutions.

## Author Contributions

**Conceptualization:** F. Antonio Medrano.

**Data curation:** F. Antonio Medrano.

**Formal analysis:** F. Antonio Medrano.

**Investigation:** F. Antonio Medrano.

**Methodology:** F. Antonio Medrano.

**Validation:** F. Antonio Medrano.

**Visualization:** F. Antonio Medrano.

**Writing – original draft:** F. Antonio Medrano.

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
