## [Decision Letter · Decision Letter 0]

25 Jan 2021

PONE-D-20-40796

Effects of Raster Terrain Representation on GIS Shortest Path Analysis

PLOS ONE

Dear Dr. Medrano,

Thank you for submitting your manuscript to PLOS ONE. After careful consideration, we feel that it has merit but does not fully meet PLOS ONE’s publication criteria as it currently stands. Therefore, we invite you to submit a revised version of the manuscript that addresses the points raised during the review process.

The referees find this work well-written, interesting, and relevant. Based on my reading, I agree. The referees recommend some minor revisions be made prior to acceptance.  In particular, they note that the intro and motivation could use a little more finesse.  They also mention a variety of relatively simple changes that could be made to strengthen the presentation of the research and more recent research on the topic that should be considered.  I’ve also listed some comments below for you to take into consideration in your revision.

Other Editor Comments:

1. Data availability statement: It’s a requirement to make the data available through a permanent data repo such as figshare or Mendeley data and the dataset should have a permanent doi.  While you can create a doi to reference work in github, it’s a more difficult process.

2. There are no citations to research appearing in PLOS journals.  Some citations should be added to PLOS manuscripts to strengthen the relationship to the journal’s readership.

3. Abstract: It may be worth mentioning a few example applications that may be familiar to the readership of PLOS that could benefit from your research findings just to provide the readers with some context.  Also, ‘GIS’ is used in the abstract, but is not defined.  Those two items should be addressed in the intro and motivation as well.

4. The key words should be terms that are not used in the title and/or abstract.  The key words are supplemental search terms that make your article more discoverable.

5. The description of the R=2 criterion and its actual implementation are not clearly aligned.  While the R=2 is explained as a Queen’s move followed by a Knights’s move in the text, the figure shows straight lines between the raster cells.  This may be confusing to the journal readership.  I’m assuming the actual movement between a cell and it’s Queen + Knight counterpart is by way of a straight line and not one following the Queen+Knight movement (a longer path).  Also, isn’t it sufficient to term it a Knight’s move?  Further, it may be worth mentioning in the manuscript exactly how arc attributes were computed for the Knight’s move (e.g., was it a sum of the cells in the Knight’s move or the sum of the portions of the cells in the straight line connecting the two cells?).

We look forward to receiving your revised manuscript.

Kind regards,

Timothy C. Matisziw, Ph.D.

Academic Editor

PLOS ONE

Journal Requirements:

2.We note that Figure(s) 3, 4, 7 and 8 in your submission contain map images which may be copyrighted. All PLOS content is published under the Creative Commons Attribution License (CC BY 4.0), which means that the manuscript, images, and Supporting Information files will be freely available online, and any third party is permitted to access, download, copy, distribute, and use these materials in any way, even commercially, with proper attribution. For these reasons, we cannot publish previously copyrighted maps or satellite images created using proprietary data, such as Google software (Google Maps, Street View, and Earth). For more information, see our copyright guidelines: http://journals.plos.org/plosone/s/licenses-and-copyright.

a)  You may seek permission from the original copyright holder of Figure(s) 3, 4, 7 and 8 to publish the content specifically under the CC BY 4.0 license. 

Reviewers' comments:

Reviewer's Responses to Questions

**Comments to the Author**

1. Is the manuscript technically sound, and do the data support the conclusions?

Reviewer #1: Yes

Reviewer #2: Yes

Reviewer #3: Yes

2. Has the statistical analysis been performed appropriately and rigorously? 

Reviewer #1: N/A

Reviewer #2: Yes

Reviewer #3: Yes

3. Have the authors made all data underlying the findings in their manuscript fully available?

Reviewer #1: Yes

Reviewer #2: Yes

Reviewer #3: Yes

4. Is the manuscript presented in an intelligible fashion and written in standard English?

Reviewer #1: Yes

Reviewer #2: Yes

Reviewer #3: Yes

5. Review Comments to the Author

Reviewer #1: The paper details representation and manipulation issues associated with the optimization of a corridor or path across space. In particular, the very much standard approach for identifying an optimal corridor/path is shown to be sensitive to potential movement in a GIS environment as well as attribute scaling. Further, the approach within the context of multi-objective optimization is shown to be further impacted.

In general, this is a fantastic topic, and of much contemporary concern. The paper is very good, but probably could be improved through further revision. Offered below are aspects of the presentation of the work that could be enhanced prior to publication:

1. The paper begins by suggesting that GIS is increasingly popular form of analytics, yet results can be sensitive to spatial representation. Then proceeds to outline the case of corridor/path analysis. Not sure this is the most effective motivation as corridor / path siting across continuous space is a fundamentally important problem on it own. Thus, either the transition to corridor analysis should be improved, or begin with corridor and make the general connection to GIS later in the section.

2. The corridor context is not one of generating a raster network, but rather that has been a common discretization approach taken to make the problem more manageable, particularly in a GIS environment. The discussion of the problem seems to suggest raster issues, but actually these are the byproduct of a selected abstraction process. This should be made more clear in the paper.

3. Similarly, multi-objective is but one approach taken, recognizing that many considerations go into corridor / path optimization. I think the MOSP introduction is a bit misleading, and sort of awkwardly brought into the paper.

4. Figure 1 seems familiar. This may well be an often used way to depict options for movement in a raster environment, but perhaps it needs a source citation.

5. Figure 2b showing only two non-dominated solutions makes it difficult to comparatively understand the a dominated solution. I realize that three paths are shown in Figure 2a, but perhaps another path should be added. I guess too that the different colors are make it hard to visualize and understand this.

6. Referencing in the paper is inconsistent. In some places, two authors are cited using an and, e.g., as Huber and Church, but in others only a comma is used, e.g., Seegmiller, Shirabe. This happens in many places, so should be cleaned up.

7. Minor editing. I saw a few things, but nothing major. These can be found by the authors in another reading of the paper, so I will not provide particular instances.

In summary, I like much of the paper, and think that the results are compelling. This should be published. However, I believe that the introduction and motivation could be enhanced through further revision.

Reviewer #2: This is an interesting article covering issues that arise when conducting shortest path analysis on raster terrains. While the article is well-written and has contribution to the field, I would like to see these changes in the manuscript.

1. In addition to network connectivity and range of attributes scale, the raster cell size is another factor that can impact both computation time and result of shortest path analysis. It would be interesting to add another dimension to current work by varying cell size of raster data and report how number of nondominated solutions, objective values, diversity of solutions and computation time change.

2. It would be good to add another table and describe 3 current networks (R1, R2 and R3) as well as new ones with different cell sizes (if any) as described in comment #1. Name and report number of nodes, arcs, cell size, … for each network.

3. There are new and closer works to the scope of the article and journal that can be used in the introduction section when describing multiobjective shortest path models. I have included two of them here:

“Matisziw TC, Gholamialam A, Trauth KM. Modeling habitat connectivity in support of multiobjective species movement: An application to amphibian habitat systems. PLoS Comput Biol. 2020; 16(12): e1008540. https://doi.org/10.1371/journal. pcbi.1008540.”

“Gholamialam A, Matisziw TC. Modeling bikeability of urban systems. Geogr Anal. 2019; 51(1):73–89.”

4. Provide legend for raster data in Fig. 3. Land use/land cover type (a), range of variables for slope (b) and costs (c and d).

5. In Fig. 4, use letters a, b and c for each panel (similar to Fig. 3) and use in the caption. Update R1, R2 and R3 with network names described in comment #2 if changes were made.

6. Use a, b, … letters for each panel in Fig. 7 and use them in the caption. Also, move range and number of solutions for each panel to the caption and refer to them by the letters.

7. It seems like 15 biobjective shortest path models have been experimented in this research. Three of which for different connectivity schemes (R1, R2 and R3), six models for fixed minimum range and six models for range shift. I highly recommend to report computation time for every single experiment. This can be worked into a new table describing range, computation time, network name and connectivity method (R1, R2 and R3). If possible, this table can be merged with the new table described in comment #2.

8. There are many ways to summarize objective values of multiobjective shortest paths. Some good examples can be found in two new references mentioned in comment #3. Try and summarize path objective values for each model and explain the variations among objectives and different models. For each experiment, the average, standard deviation, and range of objectives for each supported nondominated solution set can be easily reported.

9. Page 1, abstract, line 4. The word “on” is missing in the sentence. It should be: “Users should understand the impacts that data representations may have on their results in order to prevent distortions in their outcomes.”

10. Page 5, line 8 and 9. Replace decision space with objective space.

11. Page 8, line 5-9. How that three or four percent change has been measured. That should be first reported in a table, a figure or a graph so the reader knows how those numbers have been calculated.

12. Please consider numbering lines in your revised word document so reviewers can point to the text more easily.

13. Page 8, four listed approaches can be just explained in the text.

14. Table 1 seems like two tables with one caption. It should be merged or have separate captions.

15. Same comment for Table 2. See comment #14.

16. Page 11, last paragraph. The stopping criteria for [0,1] range should be clearly described instead of “whatever tie-breaking rules were used in the implementation.”

17. Once mentioned changes are made, some of the interesting quantitative variations for different raster representations should be mentioned/added to the abstract.

Reviewer #3: The manuscript studies the consequences of calculating shortest paths on raster data when two different aspects involved in the process are altered: network connectivity and range of the attribute scale to assign cost. The manuscript is very well written and easy to read. Comments follow.

- Pg. 7: please provide a short description of what the authors mean by “Qualitatively we compare..”. Even though, later it becomes clearer it is important to fully qualify what they are looking at.

- Pg. 7: the description of paths based on R=0, 1 and 2 seem obvious based on how the cells are connected to each other to form the network. Are there any other characteristics that the authors can provide besides this?

- Can the authors add a little more to the discussion section? For example, what happens with R=0 and R=1 networks? Yes, it is clear that the authors want to push for GIS software to include R=2 representations, but for completeness it would be important to see the differences on the paths based on the ranges on those networks too. Also, do the authors believe that a different algorithm for the shortest path calculation would yield different results?

- Please add a scale bar and north arrow to the maps.

6. PLOS authors have the option to publish the peer review history of their article (what does this mean?). If published, this will include your full peer review and any attached files.

Reviewer #1: No

Reviewer #2: No

Reviewer #3: No

---

## [Author Response · Author response to Decision Letter 0]

22 Mar 2021

Dear PLOS ONE Referees and Editors,

Thank you for taking the time to review this article. Your comments were extensive, and enlightening to many items that needed clarification and correction in the first draft of this paper. In this revision I have addressed the reviewer comments, and in addition to the article submission I have provided a copy with annotated changes between the first draft and the current revision.

Editor Comments

1. I have created a DOI reference to a snapshot of the Github repository, which can be found here: https://doi.org/10.5281/zenodo.4540743, and included it in the article.

2. I have added 4 PLOS article references, all of which provide strong support to this article. PLOS journals contain an excellent body of literature, thank you for suggesting such a strong reference resource. [23], [39-41].

3. GIS is now defined in the abstract and introduction. I have also added one application in the abstract to provide a concrete example that the reader can connect with.

4. I have updated the keywords to enable better discoverability. Thank you.

5. Good point, I have revised the connectivity definitions to be more well-defined, and have added a reference for additional clarification.

Journal Requirements

1. I have done my best to follow PLOS ONE style requirements.

2. All maps in this article were created programmatically by myself, the author. The raw data came in the form of numerical matrices, then rendering was performed with Java and the Processing API. All data and code used to generate the maps are available in the open-source Github repository. As the author and owner of all images in this publication, I grant PLOS ONE full rights to publish the images in this article.

 

Reviewer 1 Response

Thank you for this review, you bring up some excellent points that were very important to address. My comments are as follows:

1. I agree that the article made an abrupt transition from general GIS analysis to talking about corridor location. I have addressed this and improved the transition from the former to the latter in the introduction.

2. Indeed, the article needed a word about how representing continuous space on a digital computer requires discretization of both space and attributes, and that raster is used for corridor location since spatial information often originates from remote sensing imagery. I have added this to the background.

3. Thank you, the transition to MOSP was indeed awkward, I have revised the section introducing multi-objective analysis to do so more smoothly.

4. While the general content of Figure 1 is common to many publications, the exact figure was created by the author from scratch using PowerPoint.

5. I have revised the representative figure to add more solutions in objective space. This, along with the later figures in the article, should clarify the relationship between decision space and objective space, and how MOSP problems have many solutions to consider.

6. & 7. I have cleaned up references and other punctuation. Thank you for pointing this out.

Reviewer 2 Response

You bring up many good points, and you clearly did a careful and methodical review. Thank you for the time you have invested, my comments are as follows:

1. Indeed, cell size can have a major impact on spatial analysis. This has been well studied in the literature before, references [36] and [37] specifically looked at the ramifications of cell size on shortest path analysis. I focus this article on aspects that are not already prevalent in the literature.

2. This paper is quite long as-is, but this is an excellent topic for a follow-up study. Thank you for this great idea!

3. These are excellent references, thank you for pointing them out. I have added them to this article [23] & [24].

4. All maps in Fig 3 were created using Java for this study, not with GIS software, so a graphical legend is not a built-in capability. Fig 3a though uses the same colors as the NLCD2016 class legend, and the other maps in the figure use light colors represent low slope or cost, and dark colors to represent high slope or cost according to the classifications listed in Table 1. I have provided a web link to the NLCD2016 class legend in the article, and added a reference from Fig 3 to Table 1 as well.

5. Fixed.

6. I have added letters to both Fig. 7 and Fig 8, and have updated figure captions accordingly. I believe the legends should remain within the panels for better visual association.

7. This paper is quite long as-is, so I focused on analytical results rather than computation times. Also, I have previously published another study focusing only on computation times using the same data and algorithms, you are invited to read that at [49].

8. Indeed, the referenced articles do a thorough quantitative analysis of the path results. Here the focus is primarily on the qualitative differences in the analysis, but will certainly use the same techniques from the referenced publications when we do a follow-up quantitative analysis.

9. Thank you, it’s now fixed.

10. Great catch, thank you!

11. That was a preliminary rough calculation, and without the associated thorough analysis should not be included. It has been removed. Thank you for pointing this out.

12. Done.

13. We tried explaining them in paragraph form, and it was hard and confusing to follow which was which. We believe it is much clearer to enumerate them in list format.

14. & 15. I have merged the tables, and am very happy with the result. Thank you for pointing this out.

16. This was poor wording on our part, this has been amended. Thank you.

17. We certainly will do so in our follow-up quantitative study.

Reviewer 3 Response

Thank you for your review comments. They are very well focused on important points, and my comments are as follows:

1. Thank you, this needed additional clarification. This has been amended.

2. Agreed, the analysis needed a conclusion to tie-together the observations. We have added this.

3. Indeed, we did run the analysis for all combinations of classification ranges and radius values. I have addressed this in the discussion now. For the sake of brevity, I do not include figures for all 33 combinations in the article, but I have added a folder to the Github repository containing image captures of all analysis results, and of course a reader could replicate the results with the code and data posted on Github.

Regarding different results with different shortest path algorithms, any exact shortest path algorithm should give the same result if there is only one optimal path for the given data. In the rare case that there is more than one optimal path, the objective values for all optimal paths would be the same, but the decision space route might vary with different algorithms. The only instance where this would not be a rarity would be if the lower bound on the classification is zero, which is a pathological, un-realistic scenario in real-world applications.

4. The figures were created programmatically with Java and the Processing API, not GIS software, and thus scale bars and north arrows are not built-in functionality. But scale and orientation are very important items to note, so I have added clarification to the Data section as it applies to all maps in this article. Thank you for this observation.

To all reviewers and editors:

Thank you again all for your reviews and comments, I firmly believe this article is much stronger now thanks to your contributions.

F. Antonio Medrano

Texas A&M University–Corpus Christi

---

## [Editor Report · Decision Letter 1]

31 Mar 2021

Effects of Raster Terrain Representation on GIS Shortest Path Analysis

PONE-D-20-40796R1

Dear Dr. Medrano,

We’re pleased to inform you that your manuscript has been judged scientifically suitable for publication and will be formally accepted for publication once it meets all outstanding technical requirements.

Kind regards,

Timothy C. Matisziw, Ph.D.

Academic Editor

PLOS ONE

Additional Editor Comments (optional):

All of the minor referee remarks appear to have been sufficiently addressed - Thank You!
---

## [Editor Report · Acceptance letter]

5 Apr 2021

PONE-D-20-40796R1 

Effects of raster terrain representation on GIS shortest path analysis 

Dear Dr. Medrano:

I'm pleased to inform you that your manuscript has been deemed suitable for publication in PLOS ONE. Congratulations! Your manuscript is now with our production department. 

Kind regards, 

on behalf of

Dr. Timothy C. Matisziw 

Academic Editor

PLOS ONE